# The Characterization of Slovinky Sludge Bed Material Using Spectroscopic Methods

Lubica Kozakova, Maria Kanuchova *, Tomas Bakalar and Henrieta Pavolova

Faculty of Mining, Ecology, Process Control and Geotechnologies, Technical University of Kosice, Letna 9, 04200 Kosice, Slovakia; lubica.kozakova@tuke.sk (L.K.); tomas.bakalar@tuke.sk (T.B.); henrieta.pavolova@tuke.sk (H.P.)
* Correspondence: maria.kanuchova@tuke.sk; Tel.: +421-055-602-2990

**Abstract:** Slovakia has a long and distinguished tradition in the field of mining and in the processing of raw materials such as gold, silver, copper, and iron ores. In medieval times, the area that is today's Slovakia was one of the most important producers and processors of sulfide ore, which was processed specifically by flotation. Flotation waste is the remains of fine-grained materials that are deposited in sludge beds after mining and mineral processing activities. Flotation waste also contains residues of heavy metals, which can pose a potential risk to the surrounding environment. The areas with this deposited material (heaps and sludge beds) have been classified as an environmental threat and require regular monitoring by government bodies. The Slovinky sludge bed is one of these areas. The aim of the work was therefore to investigate the selected physico-chemical properties of sludge flotation waste using spectroscopic methods (XPS, XRF, and AAS). Our findings showed that the concentrations of Cu, Zn, and Ni exceeded the limits set by the relevant legislation by several fold even two decades after the end of mining and processing activities. Although the sludge bed material is alkaline, our results showed that the sludge bed material could be a potential source of selected heavy metals. The obtained data could help in the protection and restoration of areas affected by the mining and processing of sulfide ore.

**Keywords:** toxic elements; sludge bed material; chemical analysis; environmental burden

## 1. Introduction

Almost every product and service in the modern world relies on the raw materials generated through mining and mineral processing, and they have contributed significantly to the progress of human civilization and national economies. However, these activities may have a potentially serious impact on the environment [1].

Mining, especially the mining of gold and silver ores, is the best-known and most valuable contribution of Slovakia to European history. The history of the current territory of the Slovak Republic is closely connected with the prospecting for and extraction of these metals. In the past, Slovak mining was a prominent feature of Slovak spiritual and material history. The perspective on mining changed in the early 1990s and is linked to the social changes undergone by society.

In the regions with a historical tradition of mining (central Slovakia, Little Carpathians, etc.), sympathy for mining was supported by tradition and certain material advantages. After 1989, intensive campaigns by various environmental organizations began to emerge, and public opinion gradually began to oppose mining. During the period of totalitarianism, these movements were basically the only ones that could, to some extent, criticize the shortcomings of the totalitarian regime regarding the environment, which also contributed to the authority of these organizations. Several areas of Slovakia are affected by the mining of mineral resources and the processing of extracted or imported minerals [2]. Old environmental burdens remain after these activities ceased. Environmental burdens are

areas contaminated by various industrial activities. They are carried over to the present and remain a problem as an environmental debt in the sustainable development of society in the EU and in regions of Slovakia [3]. Heaps and sludge beds are remnants from mining and mineral processing activities. The Slovinsky sludge bed in Eastern Slovakia (Figure 1) is one of these burdens, containing flotation waste from the mineral processing of sulfide ore and the metallurgical processing of industrial slag in fine-grained form [4]. The municipality of Slovinky is known for its centuries-old history of mining. First, copper ore was mined, and later, iron ore was metallurgically processed in nearby Krompachy. The mining activity was terminated in Slovinky in 1993. The flotation sludge from the ore treatment plant was deposited in the sludge bed, which was in operation between 1968 and 1999. More than 4.8 million cubic meters of sludge are still stored here. The Slovinky landfill site is included in the State Program for Remediation of Environmental Pollution (2010–2015). The development of new technologies for the treatment of raw materials, especially flotation, made it possible to increase the efficiency of the processes applied to obtain useful components in the treatment of ore and non-ore raw materials. Chemical compounds of predominantly organic nature are used in the flotation process itself [5].

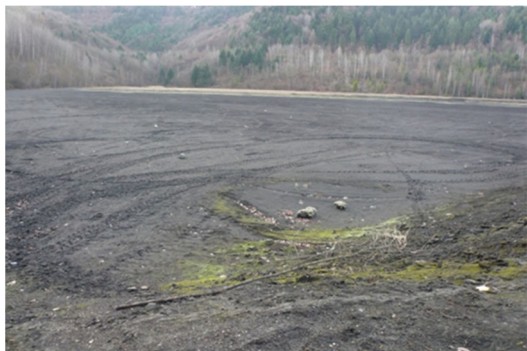 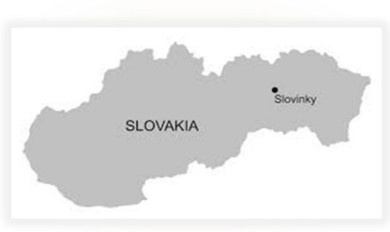

**Figure 1.** Slovinky sludge bed—view at the top (source: elaborated by authors).

The basis of flotation, as a process of mineral raw material treatment, depends on the different physico-chemical properties of the separated mineral grains' surfaces (in our case sulfide ore), which are conditioned by different specific surface energies for individual minerals. Professor K.A. Razumov explains the essence of the flotation process as follows: "Flotation, as a mineral separation process, is based on the different ability of mineral grains to attach and settle on the surface of the phase interface." The different abilities of mineral grains to stay on the surface of the phase interface results from their different specific surface energies, the size of which depends on the chemical composition and structure of the mineral lattice. Flotation separation of minerals can be carried out on the surface of the liquid–gas, liquid–liquid, liquid–solid, or gas–solid interphases [6]. During the flotation process, economically interesting Cu–Mo–sulfides are extracted, whereas pyrite and other gangue minerals are generally depressed from flotation, being exposed to oxidation by deposition at the tailings impoundment. It is generally accepted that sulfide oxidation, and in particular that of pyrite, is the main reason for the formation of acid rock drainage (ARD). These heavy-metal-loaded acid effluents are the principal environmental problem facing the mining industry today [7].

Waste from this process—flotation waste—is hydraulically transported, floated, and deposited in the sludge bed. In the alluvial process, sedimentation processes are applied depending on the granulometric composition and physical properties of the deposited material.

Larger and specifically heavier grains are deposited at the beginning of the drift path, while finer and specifically lighter grains are deposited further into the center of the sludge bed. The material washed up and deposited in this way is characterized by specific properties that are mainly related to the treatment method or chemical compounds used. The volume of this waste represents millions of m$^3$. The Slovinsky sludge bed is the highest sludge bed in Slovakia (113 m from the foot of the dam). The amounts of stored

flotation sludge are estimated at 4.8 million m$^3$. The gray-to-black colored material consists of fine grains of metals and their oxidation products from the flotation separation of ore concentrates. The surface layer consists of fine particles of extracted ore [5]. In the case of torrential rains, the stability of the embankment could be disrupted and there could be the danger of an environmental disaster. Because of lack of security, the sludge bed still represents a danger to the nearby town of Krompachy [8].

The Slovinky sludge bed has been studied by several authors from different views. Petrak [9] monitored the leachability of selected potentially toxic elements from anthropogenic material at the Slovinky sludge bed. According to Laubertova and Gerhartova [3], the Slovinky sludge bed has usable metal-bearing potential and they recommended hydrometallurgical processing. However, this requires a more detailed survey of the chemical and mineralogical composition, technical construction, and qualitative and quantitative conditions. Toth et al. [4] evaluated the sediments of the Slovinky sludge bed. Findorakova et al. [10] investigated the mineralogical characteristics of the contaminated sediment in the grain fractions by thermal analysis. Angelovicova et al. [8] monitored the contents of heavy metals in the soil, the soil reaction, and the species composition of vegetation in the emission field of the sludge bed. Selected water parameters from the stream flowing out of the Slovinky sludge bed were studied by Kozakova et al. [11]. Selected characteristics of the sludge material (neutralization potential, determination of cations and anions) were examined in the work of Kozakova et al. [12]. According to Kukla et al. [13], the air of forest ecosystems near Slovinky is polluted mainly by pollutants produced by mining and mineral processing. Slesarova [14] focused on the quality of mining waters in the Slovinky locality. The monitored parameters indicated the acceleration of the oxidation process and the formation of AMD. Cech et al. [15] described the unsatisfying condition and negative influences of the sludge bed on individual parts of the country. This monitoring complemented the given studies and contributed to a comprehensive view of environmentally burdened areas.

The goal of this research was to study the selected physico-chemical properties of sludge bed material including flotation waste from sulfide ore mineral processing, especially the particle size, chemical composition, pH, electrical conductivity, total dissolved solids, and concentration of heavy metals.

## 2. Materials and Methods

The studied material was collected from the surface of the Slovinky sludge bed. It was a dark gray, fine-grained material. Most of the physico-chemical properties of the observed material were analyzed in prepared extracts. Material for the preparation of the extracts was homogenized and sieved under 1 mm (Figure 2a,b).

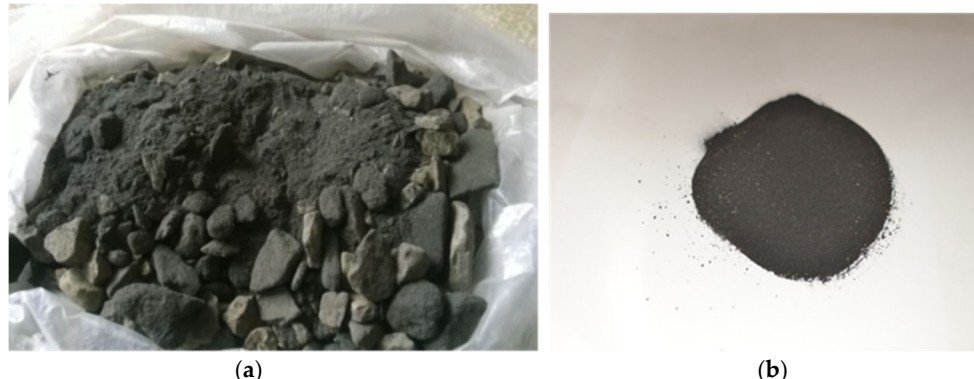

| (a) | (b) |

**Figure 2.** (**a**,**b**) The material from the top layer and homogenized, sieved material under 1 mm (source: elaborated by authors).

The electrical conductivity, pH, and total dissolved solids were determined in an extract using 20 g of the dried material, 50 mL of distilled water, and 50 mL of 1 mol·L$^{-1}$

KCl. The prepared samples were shaken for 2 and 24 h. A combined pH meter (HAMA Combo pH & EC) was used for the determination of pH, electrical conductivity, and total dissolved solids. The particle size was determined by sieve analysis. Seven sieves (800, 630, 315, 180, 80, 63, and 43 μm) were used. One representative sample of 100 g was weighed from each sieve. Sieve analysis was performed three times. The basic chemical composition of the tested solid samples was investigated by means of X-ray fluorescence analysis (XRF) using a SPECTRO iQ II (Ametek, Germany) with an SDD silicon drift detector at a resolution of 145 eV with 10,000 pulses. The primary beam was polarized using a Bragg crystal and Highly Ordered Pyrolytic Graphite (HOPG) target. The samples were measured for 300 s at voltages of 25 and 50 kV at currents of 0.5 and 1.0 mA, respectively, under a helium atmosphere using the standardized method of fundamental parameters for powder samples.

The surface chemical structure was studied by X-ray photoelectron spectroscopy (SPECS Surface Nano Analysis GmbH, Berlin, Germany). X-ray photoelectron spectroscopy (XPS) of samples was performed using a SPECS XPS instrument equipped with a PHOIBOS 100 SCD hemispherical energy analyzer and a non-monochromatic X-ray source. The survey surface spectrum was measured at 70 eV pass energy and core spectra at 20 eV at room temperature. All spectra were acquired at a basic pressure of 10-8 mbar with AlKα excitation at 10 kV (200 W). Concentrations of heavy metals in an aqueous extracts were measured by means of AAS (iCE 3300 AA (ThermoFisher Scientific, Grand Island, NY, USA)). They were determined in an extract consisting of 1 g of solid material and 25 mL of 0.5 mol·L$^{-1}$ HCl. The prepared samples were subjected to two-hour shaking in a horizontal shaker (Environmental Incubator Shaker EC–20/60—Biosan, Riga, Latvia) in wide-neck bottles.

## 3. Results and Discussion

The grain distribution of the studied material is shown in the Table 1. The size fraction 0.080 to 0.180 mm had the largest representation.

**Table 1.** The particle grain distribution of sieved material (source: elaborated by authors).

| No. | Size Fraction [mm] | Weight of Sample [g] | | | Average Values [g] | % Weight Retained [%] | Cumulative % Weight Retained [%] | Cumulative % Weight Passing [%] |
| | | 1. | 2. | 3. | | | | |
| --- | --- | --- | --- | --- | --- | --- | --- | --- |
| 1 | 0.800–1.000 | 7.42 | 8.35 | 8.95 | 8.24 | 8.24 | 8.24 | 91.76 |
| 2 | 0.630–0.800 | 9.16 | 10.46 | 9.16 | 9.59 | 1.35 | 9.59 | 90.41 |
| 3 | 0.315–0.630 | 12.32 | 11.42 | 8.73 | 10.82 | 1.23 | 10.82 | 89.18 |
| 4 | 0.180–0.315 | 10.94 | 11.94 | 13.63 | 12.17 | 1.35 | 12.17 | 87.83 |
| 5 | 0.080–0.180 | 46.3 | 45.69 | 47.2 | 46.40 | 34.23 | 46.40 | 53.60 |
| 6 | 0.063–0.080 | 65.3 | 65.75 | 66.24 | 65.76 | 19.37 | 65.76 | 34.24 |
| 7 | 0.043–0.063 | 73.23 | 74.98 | 75.99 | 74.73 | 8.97 | 74.73 | 25.27 |
| 8 | 0–0.043 | 26.77 | 25.02 | 24.01 | 25.27 | 25.27 | 100.00 | 0.00 |

The chemical composition of the solid material by means of XRF is presented in the following tables (Tables 2 and 3).

**Table 2.** The chemical composition of studied material by means of XRF [%] (source: elaborated by authors).

| As$_2$O$_3$ | CuO | Cr$_2$O$_3$ | Fe$_2$O$_3$ | ZnO | PbO | Al$_2$O$_3$ | Sb$_2$O$_5$ | SnO$_2$ | CaO | MgO | SiO$_2$ | Rest |
| --- | --- | --- | --- | --- | --- | --- | --- | --- | --- | --- | --- | --- |
| 0.001 | 0.890 | 0.644 | 36.96 | 5.772 | 0.490 | 4.826 | 0.429 | 0.278 | 6.880 | 5.955 | 27.96 | 8.916 |

**Table 3.** The concentrations of heavy metals by means of XRF and legislative limit values.

| Concentrations of Selected Heavy Metals c [mg·kg⁻¹] | | | | | | | |
|---|---|---|---|---|---|---|---|
| Cr | Ni | Cu | Zn | As | Pb | Cd | Hg |
| 4748 | 439.7 | 8803 | 33,200 | 488.8 | 4535 | 98.9 | 16.7 |
| Limit values for sediments according to the Slovak legislation [mg·kg⁻¹] | | | | | | | |
| 1000 | 300 | 1000 | 2500 | 20 | 750 | 10 | 10 |

Oxides $Fe_2O_3$ (36.96%) and $SiO_2$ (27.96%) had the highest percentage representation (Figure 3). These results corresponded with the data of Tóth et al. [4], who reported an $SiO_2$ value of 38.96% and Fe (total) value of 41.61%. The value of CaO (6.88%) also corresponded with the data of Toth et al. [16], who reported 6.93%.

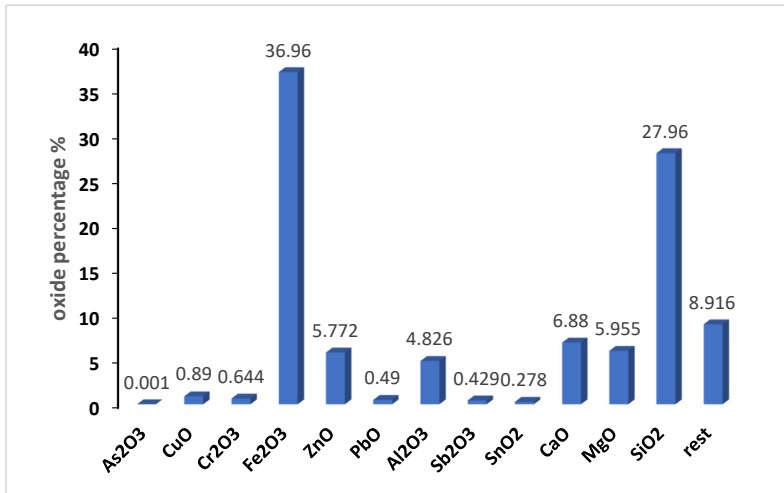

**Figure 3.** Graphic evaluation of the oxide percentage representation (source: elaborated by authors).

The measured heavy metal concentrations in the sludge bed material were compared with the limit values according to Act No. 203/2009 Coll., on the application of sewage sludge and bottom sediments to amended soil (Table 3, Figure 4).

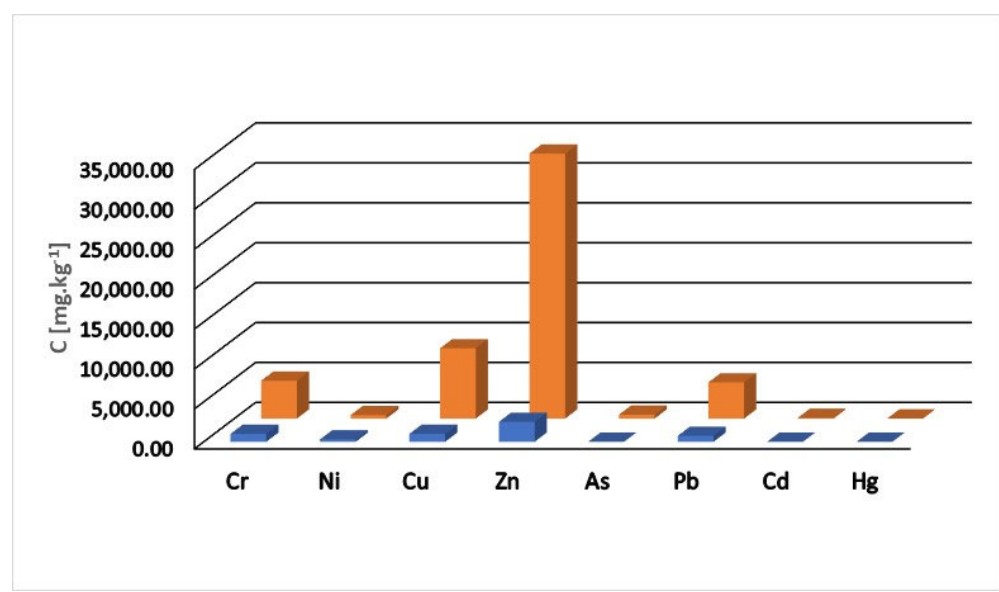

**Figure 4.** A comparison of heavy metal concentrations with limit values (blue color is the limit value) (source: elaborated by authors).

The concentrations of all selected elements, Cu, Zn, As, Pb, Cd, Cr, Ni, and Hg, far exceeded the limit values in comparison with Act No. 203/2009.

Some of the results mentioned above were comparable to the data published by Toth et al. [16]: Cu 7296 mg·kg$^{-1}$, Zn 28,251 mg·kg$^{-1}$, As 310.3 mg·kg$^{-1}$.

Analogous values for Ni, Cu, Zn, and As were also given by Findorakova et al. [10], who monitored the concentrations of heavy metals in different size fractions. In the investigated size fraction (63 to 250 μm), the values for Ni were in the range of 293.4 to 436.8 mg·kg$^{-1}$; Cu was in the range of 6275.5 to 8895.8 mg·kg$^{-1}$; Zn was in the range of 16,996.5 to 30,966.3 mg·kg$^{-1}$; and As was in the range of 327.4 to 419.6 mg·kg$^{-1}$. Petrak et al. [9] also found similar concentration values for As (443 mg·kg$^{-1}$) and Cu (8418 mg·kg$^{-1}$), but a lower value for Pb (3811 mg·kg$^{-1}$).

The electrical conductivity and total dissolved solids values after 2 and 24 h of shaking were almost identical. The pH of the extract as well as the temperature were increased slightly after 24 h of shaking (Tables 4 and 5).

**Table 4.** The evaluation of measurement data after 2 h of shaking (source: elaborated by authors).

| No. | Sample Preparation (Leachate) | Temperature [°C] | pH | EC [mS·cm$^{-1}$] | Total Dissolved Solids [ppt] |
|---|---|---|---|---|---|
| 1 | 20 g + 50 mL H$_2$O | 23 | 8.51 | 0.23 | 0.12 |
| 2 | 20 g + 50 mL H$_2$O | 23 | 8.63 | 0.3 | 0.13 |
| Average value | | 23 | 8.57 | 0.265 | 0.125 |
| 3 | 20 g + 50 mL 1 mol/L KCl | 23 | 8.68 | over 20 | over 10 |
| 4 | 20 g + 50 mL 1 mol/L KCl | 22.7 | 8.73 | over 20 | over 10 |
| Average value | | 22.85 | 8.705 | over 20 | over 10 |

**Table 5.** The evaluation of measurement data after 24 h of shaking (source: elaborated by authors).

| No. | Sample Preparation (Leachate) | Temperature [°C] | pH | EC [mS·cm$^{-1}$] | Total Dissolved Solids [ppt] |
|---|---|---|---|---|---|
| 1 | 20 g + 50 mL H$_2$O | 24.5 | 8.67 | 0.24 | 0.12 |
| 2 | 20 g + 50 mL H$_2$O | 24.5 | 8.98 | 0.23 | 0.12 |
| Average value | | 24.5 | 8.825 | 0.235 | 0.12 |
| 3 | 20 g + 50 mL 1 mol/L KCl | 24.3 | 8.87 | over 20 | over 10 |
| 4 | 20 g + 50 mL 1 mol/L KCl | 24.1 | 8.93 | over 20 | over 10 |
| Average value | | 24.2 | 8.9 | over 20 | over 10 |

The sludge bed material had an alkaline character. The pH value ranged from 8.57 to 8.9. Petrak et al. [9] published pH values of 8.1 to 8.7, while Toth et al. [4] gave pH values of 7.76 to 8.69.

The specific electrical conductivity and total dissolved solids could not be measured in leachate with KCl due to the detection limits of the apparatus used. The detection limit of the combined pH meter (HAMA Combo pH & EC, Hanna Instruments Ltd., New York, NY, USA) is 20 mS·cm$^{-1}$ for electrical conductivity and 10 ppt for total dissolved solids.

The concentrations of selected heavy metals in samples of extracts by means of AAS were compared with the limit values applicable by the Slovak legislation for hazardous waste landfills (Decree of the Slovak Republic Ministry of the Environment No. 310/2013 Coll., which implements some provisions of the Waste Act).

The heavy metals zinc, copper, and nickel exceeded the limits. Cadmium and chromium did not exceed the limit values (Table 6 and Figure 5).

**Table 6.** The concentrations of selected heavy metals by means of AAS (source: elaborated by authors).

| Concentrations of Selected Heavy Metals [mg·L$^{-1}$] | | | | | |
|---|---|---|---|---|---|
| **Cr** | **Ni** | **Cu** | **Zn** | **Fe** | **Cd** |
| 6.478 | 10.347 | 326.396 | 793.726 | 42,919.956 | 0.149 |
| Limit values for sediments according to the Slovak legislation [mg·L$^{-1}$] | | | | | |
| 7 | 4 | 10 | 20 | - | 0.5 |

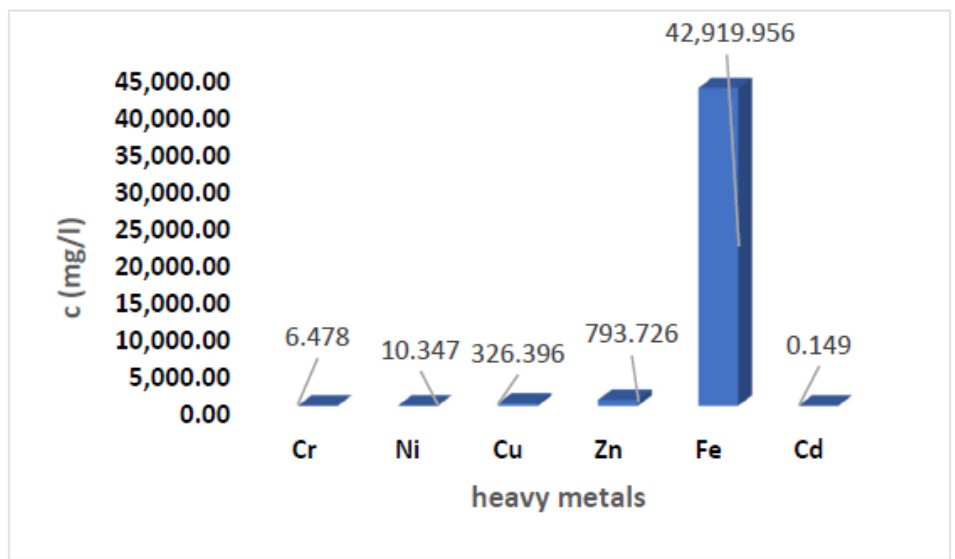

**Figure 5.** The heavy metal concentrations by means of AAS (source: elaborated by authors).

The average electrical conductivity value in the surface layer of the sludge bed reached 265 µS·cm$^{-1}$. Toth et al. [4] reported a value of 31.75 µS·cm$^{-1}$ for the surface layer material and a value of 447.29 µS·cm$^{-1}$ for the material deposited below the slag.

According to the limit values set out in the Slovak legislation (Act No. 310/2013 Coll.), the limits for heavy metals such as Ni, Cu, and Zn were found to have been exceeded. Toth et al. [4] reported increased concentrations of potentially toxic elements that may pose a potential risk of pollution to the surrounding environment, particularly Cu, Zn, Cr, Pb, Ba, Sn, As, and Sb.

The presence of heavy metals was also confirmed by XPS. In Figure 6, the survey spectrum of the sample is presented. The method did not confirm the presence of sulfide particles in the examined samples, which was in accordance with the results of XRF. It can be assumed that the sulfide particles were eliminated by the flotation process.

A method for the study of the chemical structure of the surface layers of the material—XPS analysis—evidenced the presence of the valence states for chromium. The detailed XPS spectral lines of Cr 2p core levels are shown in Figure 7.

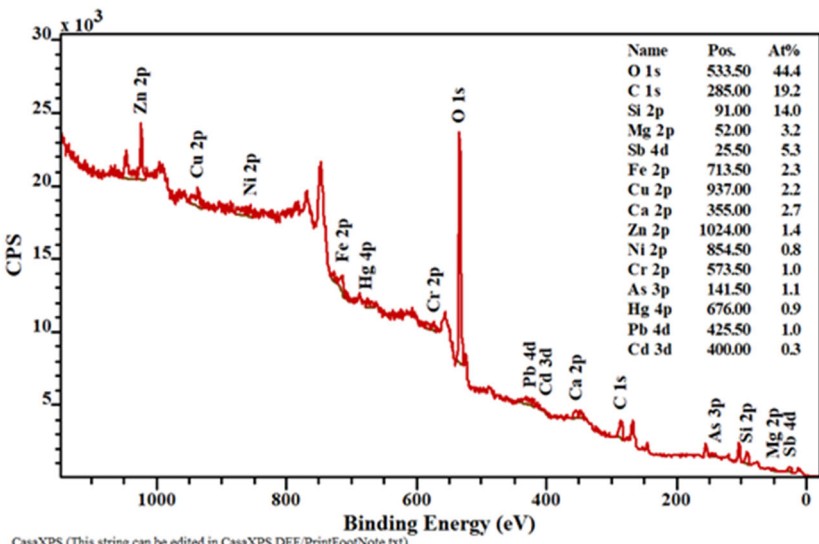

**Figure 6.** The confirmation of heavy metals by means of XPS (source: elaborated by authors).

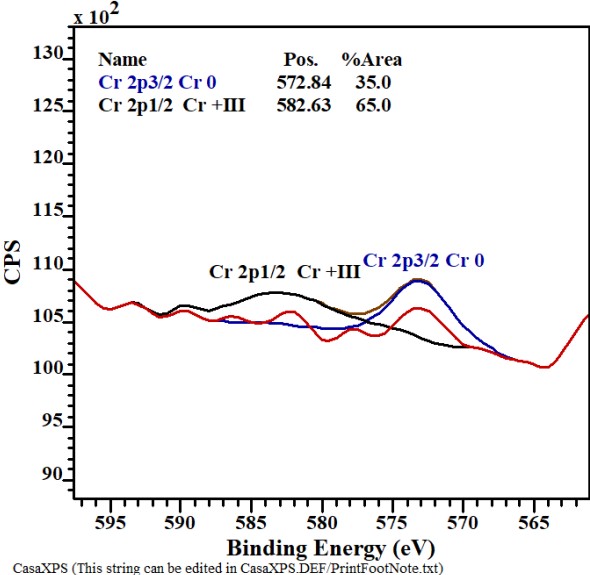

**Figure 7.** The core level spectra of Cr by means of XPS (source: elaborated by authors).

The binding energy (BE) values of 572.84 and 582.63 eV correspond to the Cr 2p3/2 and Cr 2p3/2 spectral lines of chromium in oxidation states 0 and +III, respectively. These values corresponded to species in ref [17]. Hexavalent chromium (Cr + VI) is highly toxic and mutagenic. Ingestion of chromium (VI) in water has been linked to stomach tumors and may also cause allergic contact dermatitis (ACD) [18,19]. From this point of view, it is good that the presence of Cr +VI was not confirmed.

The measured increased contents of heavy metals (especially Cu and Zn) in the surroundings of the sludge bed pointed to the fact that these metals (their highest concentrations were found in the sludge bed material) are carried into the surrounding environment by the wind and, due to other meteorological factors, subsequently reach the soil. However, the detected low values of soil reaction–pH in the vicinity of the sludge bed may cause their further mobility.

This work tried to comprehensively characterize the location, which was influenced by mining and processing activities for centuries. The site was investigated 15–20 years after these activities ceased.

Information on the state of individual components of the environment can serve as an aid in the rehabilitation of old environmental burdens, including the remains from mining

and processing activities. The monitoring of environmental burdens is currently receiving considerable attention by the relevant state authorities.

The acquired knowledge can be applied in localities where sulfide ore was mined. The work can help in the overall characterization of the Spiš region, affected by centuries of mining, refining, and metallurgical activities. This region therefore requires special attention from the point of environmental monitoring. Information obtained through environmental monitoring serves as a basis for making economic, political, and social decisions.

From the point of view of scientific focus, the work can be understood as a point of contact between technical disciplines, natural science disciplines, and ecology and environmental studies.

The purpose of this study was focused on environmental monitoring; however, the contents of selected elements (copper, iron, and nickel) were so high that it might be worth recovering them. Copper and nickel belong to the Strategy Raw Materials and are proposed for the Critical Raw Materials list 2023 [20]. One of the strategic goals in the field of raw materials is the highest possible self-sufficiency of individual countries in obtaining critical raw materials, which is related to the use of materials from old environmental burdens.

## 4. Conclusions

The aim of this study was to examine the selected physico-chemical properties of the upper layer material from the Slovinky sludge bed. The Slovinky location is included in the registry of environmental loads with high priority for monitoring.

The material deposited on the sludge bed can be a source of heavy metals, it can affect the pH of soils, and the water flowing from the sludge bed can threaten the quality of the surrounding surface and underground waters. The construction and operation of sewage treatment plants affect the country's water regime.

The gray-to-black color of the sludge bed material consisted of fine grains of metals and their oxidation products originating from the flotation separation of the ore concentrate. The surface layer of the sludge bed was made of fine-grained material originating from Kovohuty Krompachy. In this locality, even two decades after the end of mining and processing activities, there were increased concentrations of heavy metals in the soil. In addition to mining and processing activities, this may also be a consequence of the region's natural geochemical anomalies.

Based on the results of the experimental part of the work, the concentrations of all heavy metals measured in the sludge bed material significantly exceeded the limit values according to the relevant legislation. Our findings correspond with the results of other authors. The limit values of Cu and Zn were exceeded in most samples, the limit value for Ni was exceeded only in one sample, and slightly increased concentrations of Pb (compared to limit values) were recorded in only two samples.

As mentioned at the end of the discussion, the high contents of the selected metals in the samples indicate the possibility of using the sludge bed material as a potential source for obtaining selected metals.

**Author Contributions:** Each author (L.K., M.K., T.B. and H.P.) has contributed equally to this publication. Conceptualization, L.K. and M.K.; methodology, L.K., M.K. and T.B.; validation, M.K., L.K. and H.P.; formal analysis, T.B. and H.P.; resources, L.K., M.K. and H.P.; data curation, L.K., M.K. and T.B.; writing—original draft preparation, L.K. and M.K.; writing—review and editing, T.B. and H.P.; visualization, L.K., M.K. and H.P.; supervision, L.K. and M.K.; project administration, L.K. and M.K.; funding acquisition, L.K. and M.K. All authors have read and agreed to the published version of the manuscript.

**Funding:** This work is supported by the Scientific Grant Agency of the Ministry of Education, Science, Research, and Sport of the Slovak Republic and the Slovak Academy Sciences as part of the research project VEGA 1/0247/23.

**Institutional Review Board Statement:** Not applicable.

**Informed Consent Statement:** Not applicable.

**Data Availability Statement:** The data presented in this article are available on request from the corresponding author.

**Acknowledgments:** The authors would like to thank the anonymous referees for their valuable comments that improved the quality of the manuscript.

**Conflicts of Interest:** The authors declare no conflict of interest.

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
