# Peer review of "The Characterization of Slovinky Sludge Bed Material Using Spectroscopic Methods"

_sustainability, doi:10.3390/su15107887_

Round 1

Reviewer 1 Report

In the manuscript: The characterization of the material from the environmental burden Slovinky using spectroscopy methods the Authors was draw out interesting problems due to ecology and environmental matters. The results are important, interesting in many aspects and  could give some new information of the post flotation tailing in investigated area. Unfortunately the text is unclear and difficult for readers because of multiply linguistic errors.

The methods are properly used although methodology is not clear described. The Discussion  is lacking of reliable discussion of results in relation to earlier papers. Conclusion including more discuss then most important thesis if the work. Problematic is English language and terminology used in the context of tailings and sludges, what partly or completely changing meaning of the text as well as  undermined scientific data.

In my opinion the text have been not properly prepared and cannot be published in present form.

Detailed comments:

1.      The title is unclear, partly misunderstanding: the material from the environmental burden Slovinky is not informative

2.      The abstract should be rewritten , there is lacking of most important thesis of the work

3.      Historical part – should be shortened, that is not necessary to the paper

4.      LINE 76  :   properties of the separated mineral grains surface – which kind of mineral grains? 

5.      metals grains, minerals or metals have been deposited   -  not explained, what kind of components was investigated in this work, what material commonly used in the text does mean

6.      LINE 141  mesh scale is not used in millimeters or micrometers, as is in the results, should be unify/normalized

7.      LINE 169 The size distribution better   grain -  size

8.      LINE 172 particle size distribution of sieved material;  better:   grain -  size

9.      XRF method presents chemical composition in oxides for major components of sediments (e.g. Si, Al, Ca), why the minor components (Cu, Zn, Sb, As) was presented in oxides? Were there described any sulphide particle in the sludge? This problem should be discussed.

10.   Figure 3 presents result of XRF data, but the big differences with the values make results  visibility almost impossible, should be shown in different diagrams/scales

11.   Has the Act No. 203/2009 Coll. of Laws application to the wastes, or to the soils? There is not clear explained.  of sewage sludge and bottom sediments to the soil as amended (Table 3, Figure 4).

12.   LINES 207-215 that’s are discussion of results rather and should be put in Chpt. Discussion, or The title should be changed on: Results and discussion

13.   LINE 255 what exactly means: The concentrations of selected heavy metals in samples of extracts?

14.   LINES 263-265 – average electrical conductivity is exactly average values of Toth [4] which reported the value of 31.75 μS.cm -1  in the surface layer material and 447.29 μS.cm -1      in the material deposited below the slag: this is possible, that the mixed sediments from the tailing and from the basement have been measured? 

15.   material and 447.29 μS.cm -1      in the material deposited below the slag.  

16.   LINES 299-301 -there are not necessary discussing of Chromium IV insalubrity, while the core level  295  spectra of Cr by XPS method did now showed Cr IV

17.   Figures 3 and 5 are not in good quality

18.   Results, discussion and conclusion chapters should be rewrite, because of missing of the contents (some part discussion of results are included  in conclusions).

19.   Conclusions are too extend , short most important thesis of work should be done.

20.   Some of more scientific – soundness literature position is needed, connected with the tailings characteristics: (e.g. Dold, Fondbote)

Author Response

 Thank you very much for the valuable advice and comments that helped us improve the quality of our article.

The title is unclear, partly misunderstanding: the material from the environmental burden Slovinky is not informative The title was changed in the article

  1. The abstract should be rewritten , there is lacking of most important thesis of the work Abstract was rewritten
  2. Historical part – should be shortened, that is not necessary to the paper this part was shortened
  3. LINE 76  :   properties of the separated mineral grains surface – which kind of mineral grains? Sulphide ores
  4. metals grains, minerals or metals have been deposited   -  not explained, what kind of components was investigated in this work, what material commonly used in the text does mean - sludge bed materialflotation waste from suphide ores mineral processing
  5. LINE 141  mesh scale is not used in millimeters or micrometers, as is in the results, should be unify/normalized It was corrected in the article
  6. LINE 169 The size distribution better   grain -  size - It was corrected in the article
  7. LINE 172 particle size distribution of sieved material;  better:   grain -  size It was corrected in the article
  8. XRF method presents chemical composition in oxides for major components of sediments (e.g. Si, Al, Ca), why the minor components (Cu, Zn, Sb, As) was presented in oxides? Were there described any sulphide particle in the sludge? This problem should be discussed. The XPS method did not confirm the presence of sulfide particles in the examined material samples, which is in accordance with the results of the XRF analytical method. We assume that the sulfide particles were eliminated by the flotation process.
  9. Figure 3 presents result of XRF data, but the big differences with the values make results  visibility almost impossible, should be shown in different diagrams/scales Figure 3 was corrected
  10. Has the Act No. 203/2009 Coll. of Laws application to the wastes, or to the soils? There is not clear explained.  of sewage sludge and bottom sediments to the soil as amended (Table 3, Figure 4). The text of the law was supplemented (Act No. 203/2009 Coll. of Laws, on the application of sewage sludge and bottom sediments to the soil as amended)
  11. LINES 207-215 that’s are discussion of results rather and should be put in Chpt. Discussion, or The title should be changed on: Results and discussion - we combined chapter 3 and 4 to Results and discussion
  12. LINE 255 what exactly means: The concentrations of selected heavy metals in samples of extracts? This is a methodological procedure for the determination of heavy metals using the AAS method - soil samples were extracted with nitric acid and hydrochloric acid
  13. LINES 263-265 – average electrical conductivity is exactly average values of Toth [4] which reported the value of 31.75 μS.cm -1  in the surface layer material and 447.29 μS.cm -1      in the material deposited below the slag: this is possible, that the mixed sediments from the tailing and from the basement have been measured?  Maybe yes, it is possible, but we do not know because our samples were taken from the sludge bed upper layer
  14. material and 447.29 μS.cm -1      in the material deposited below the slag.  It is a comparison of the results
  15. LINES 299-301 -there are not necessary discussing of Chromium IV insalubrity, while the core level  295  spectra of Cr by XPS method did now showed Cr IV
  16. Figures 3 and 5 are not in good quality figures were changed
  17. 18.Results, discussion and conclusion chapters should be rewrite, because of missing of the contents (some part discussion of results are included  in conclusions). Results, discussion and conclusion chapters were corected
  18. Conclusions are too extend , short most important thesis of work should be done.Conclusion was corrected
  19. Some of more scientific – soundness literature position is needed, connected with the tailings characteristics: (e.g. Dold, Fondbote) Thank you very much for your comments, we added these authors citation, it is very interesting, thanks again

Reviewer 2 Report

The article is interesting. However, it needs to be corrected and improved to make it more clear. The comments I presented below.

 1.      The purpose of the article is not specified. Why was the research conducted? Was the research conducted to determine the impact of elements on the environment, or maybe to determine the possibility of recovering elements? This is not clearly defined in the introduction.

2.      The contents of some elements exceed the permissible values. What is the conclusion? The Authors only mention the potential pollution of the environment and the monitoring of pollution. However, there is no information on the possibilities of limiting the migration of elements in the environment.

3.      There is no information in the discussion chapter about the use of the results. Are the content of elements so high that it would be worth recovering them? Or maybe their values are only increased enough to cause environmental pollution, soil and water pollution. In this case, it will be necessary to take the activity to reduce the content of these elements in the environment. However, the Authors, apart from the proposals of environmental monitoring, which, according to the information provided in the article, are being carried out, do not have any other proposals for eliminating harmful impact of elements for the environment.

4.      The statement ending the Conclusion Chapter, that the results of XRF tests showed that the analyzed material is slag is very general. There is no information about the type of this slag and its possible practical application in any branch of industry.

5.      The research results presented in the tables and figures are very valuable, but some fragments of the manuscript are very general, because there is no information about the possibility of using the results in practice. So that the manuscript is rather the review than the article.

 I think that the manuscript needs some corrections to make it clear. Authors should specify some information.

Author Response

Thank you very much for the valuable advice and comments that helped us improve the quality of our article.

The article is interesting. However, it needs to be corrected and improved to make it more clear. The comments I presented below.

  1. The purpose of the article is not specified. Why was the research conducted? Was the research conducted to determine the impact of elements on the environment, or maybe to determine the possibility of recovering elements? This is not clearly defined in the introduction.

Monitoring in our work could complement the given studies and contribute to a comprehensive picture of environmentally burdened areas. The introduction was extended.

  1. The contents of some elements exceed the permissible values. What is the conclusion? The Authors only mention the potential pollution of the environment and the monitoring of pollution. However, there is no information on the possibilities of limiting the migration of elements in the environment.

It was not the subject of our investigation

  1. There is no information in the discussion chapter about the use of the results. Are the content of elements so high that it would be worth recovering them? Or maybe their values are only increased enough to cause environmental pollution, soil and water pollution. In this case, it will be necessary to take the activity to reduce the content of these elements in the environment. However, the Authors, apart from the proposals of environmental monitoring, which, according to the information provided in the article, are being carried out, do not have any other proposals for eliminating harmful impact of elements for the environment.

The aim of our study was focused to an environmental monitoring and yes, the content of selected elements is so high that it should be worth to recover them,  it will be subject of interest of  the next investigation.

  1. The statement ending the Conclusion Chapter, that the results of XRF tests showed that the analyzed material is slag is very general. There is no information about the type of this slag and its possible practical application in any branch of industry. Thank you very much for the reminder, you're right, our statement is very general, so we removed that sentence. Based on the chemical composition using the XRF method, it is not possible to clearly determine whether it is flotation waste or slag.
  2. The research results presented in the tables and figures are very valuable, but some fragments of the manuscript are very general, because there is no information about the possibility of using the results in practice. So that the manuscript is rather the review than the article. We added the possibilities in practice in the discussion.

Reviewer 3 Report

The manuscript aims at the physicochemical characterization of sludge bed material Slovinsky since this sludge is rich in commercial metals. This sludge is part of contaminated sites formed by nature since the medieval edge. The manuscript contribution deals to give information to use in the rehabilitation of old environmental burdens, which also includes remains from mining and processing activities.

The manuscript falls in the scope of the journal, the information proportioned will be helpful to improve the rehabilitation of contaminated sites that contains heavy metal that has an aggregated value. However, the manuscript is subject to some changes.

1. The abstract must be rewritten considering the highlighting of the findings.

2. The discussion section must be improved, e.g. expanding explanations.

3. The quality of Fig. 3 must be improved.

4. Tables must be presented according to the guidelines of the journal.

5. Review the English grammar throughout the manuscript.

The manuscript can be considered publishable after making minor corrections.

Author Response

Thank you very much for the valuable advice and comments that helped us improve the quality of our post.

1. The abstract must be rewritten considering the highlighting of the findings. It was corrected

2. The discussion section must be improved, e.g. expanding explanations. It was corrected

3. The quality of Fig. 3 must be improved. It was corrected

4. Tables must be presented according to the guidelines of the journal. It was corrected

5. Review the English grammar throughout the manuscript. It was checked

The manuscript can be considered publishable after making minor corrections.

Round 2

Reviewer 1 Report

Dear Authors, the paper has been corrected sufficiently, due to methodology, its structure and clarity, some mineralogical explanations and new literature in bibligraphy chapter was added, language was modified, the new title and abstract are more informative then in the first version, thus manuscript can be published in present form.

Author Response

Dear reviewer,

thank you very much for your comment to improve English language. The article was corrected by English native scientist .
